# Variational Gaussian Process State-Space Models

**Roger Frigola, Yutian Chen and Carl E. Rasmussen**
Department of Engineering
University of Cambridge
`{rf342,yc373,cer54}@cam.ac.uk`

## Abstract

State-space models have been successfully used for more than fifty years in different areas of science and engineering. We present a procedure for efficient variational Bayesian learning of nonlinear state-space models based on sparse Gaussian processes. The result of learning is a tractable posterior over nonlinear dynamical systems. In comparison to conventional parametric models, we offer the possibility to straightforwardly trade off model capacity and computational cost whilst avoiding overfitting. Our main algorithm uses a hybrid inference approach combining variational Bayes and sequential Monte Carlo. We also present stochastic variational inference and online learning approaches for fast learning with long time series.

## 1  Introduction

State-space models (SSMs) are a widely used class of models that have found success in applications as diverse as robotics, ecology, finance and neuroscience (see, e.g., Brown et al. [3]). State-space models generalize other popular time series models such as linear and nonlinear auto-regressive models: (N)ARX, (N)ARMA, (G)ARCH, etc. [21].

In this article we focus on Bayesian learning of nonparametric nonlinear state-space models. In particular, we use sparse Gaussian processes (GPs) [19] as a convenient method to encode general assumptions about the dynamical system such as continuity or smoothness. In contrast to conventional parametric methods, we allow the user to easily trade off model capacity and computation time. Moreover, we present a variational training procedure that allows very complex models to be learned without risk of overfitting.

Our variational formulation leads to a tractable approximate *posterior over nonlinear dynamical systems*. This approximate posterior can be used to compute fast probabilistic predictions of future trajectories of the dynamical system. The computational complexity of our learning approach is linear in the length of the time series. This is possible thanks to the use of variational sparse GPs [22] which lead to a smoothing problem for the latent state trajectory in a simpler auxiliary dynamical system. Smoothing in this auxiliary system can be carried out with any conventional technique (e.g. sequential Monte Carlo). In addition, we present a stochastic variational inference procedure [10] to accelerate learning for long time series and we also present an online learning scheme.

This work is useful in situations where: 1) it is important to know how uncertain future predictions are, 2) there is not enough knowledge about the underlying nonlinear dynamical system to create a principled parametric model, and 3) it is necessary to have an explicit model that can be used to simulate the dynamical system into the future. These conditions arise often in engineering and finance. For instance, consider an autonomous aircraft adapting its flight control when carrying a large external load of unknown weight and aerodynamic characteristics. A model of the nonlinear dynamics of the new system can be very useful in order to automatically adapt the control strategy. When few data points are available, there is high uncertainty about the dynamics. In this situation,

a model that quantifies its uncertainty can be used to synthesize control laws that avoid the risks of overconfidence.

The problem of learning flexible models of nonlinear dynamical systems has been tackled from multiple perspectives. Ghahramani and Roweis [9] presented a maximum likelihood approach to learn nonlinear SSMs based on radial basis functions. This work was later extended by using a parameterized Gaussian process point of view and developing tailored filtering algorithms [6, 7, 23]. Approximate Bayesian learning has also been developed for parameterized nonlinear SSMs [5, 24].

Wang et al. [25] modeled the nonlinear functions in SSMs using Gaussian processes (GP-SSMs) and found a MAP estimate of the latent variables and hyperparameters. Their approach preserved the nonparametric properties of Gaussian processes. Despite using MAP learning over state trajectories, overfitting was not an issue since it was applied in a dimensionality reduction context where the latent space of the SSM was much smaller than the observation space. In a similar vein, [4, 12] presented a hierarchical Gaussian process model that could model linear dynamics and nonlinear mappings from latent states to observations. More recently, Frigola et al. [8] learned GP-SSMs in a fully Bayesian manner by employing particle MCMC methods to sample from the smoothing distribution. However, their approach led to predictions with a computational cost proportional to the length of the time series.

In the rest of this article, we present an approach to variational Bayesian learning of flexible non-linear state-space models which leads to a simple representation of the posterior over nonlinear dynamical systems and results in predictions having a low computational complexity.

## 2 Gaussian Process State-Space Models

We consider discrete-time nonlinear state-space models built with deterministic functions and additive noise

$$\mathbf{x}_{t+1} = f(\mathbf{x}_t) + \mathbf{v}_t, \tag{1a}$$

$$\mathbf{y}_t = g(\mathbf{x}_t) + \mathbf{e}_t. \tag{1b}$$

The dynamics of the system are defined by the state transition function $f(\mathbf{x}_t)$ and independent additive noise $\mathbf{v}_t$ (process noise). The states $\mathbf{x}_t \in \mathbb{R}^D$ are latent variables such that all future variables are conditionally independent on the past given the present state. Observations $\mathbf{y}_t \in \mathbb{R}^E$ are linked to the state via another deterministic function $g(\mathbf{x}_t)$ and independent additive noise $\mathbf{e}_t$ (observation noise). State-space models are stochastic dynamical processes that are useful to model time series $\mathbf{y} \triangleq \{\mathbf{y}_1, ..., \mathbf{y}_T\}$. The deterministic functions in (1) can also take external known inputs (such as control signals) as an argument but, for conciseness, we will omit those in our notation.

A traditional approach to learn $f$ and $g$ is to restrict them to a family of parametric functions. This is particularly appropriate when the dynamical system is very well understood, e.g. orbital mechanics of a spacecraft. However, in many applications, it is difficult to specify a class of parametric models that can provide both the ability to model complex functions and resistance to overfitting thanks to an easy to specify prior or regularizer. Gaussian processes do have these properties: they can represent functions of arbitrary complexity and provide a straightforward way to specify assumptions about those unknown functions, e.g. smoothness. In the light of this, it is natural to place Gaussian process priors over both $f$ and $g$ [25]. However, the extreme flexibility of the two Gaussian processes leads to severe nonidentifiability and strong correlations between the posteriors of the two unknown functions. In the rest of this paper we will focus on a model with a GP prior over the transition function and a parametric likelihood. However, our variational formulation can also be applied to the double GP case (see supplementary material).

A probabilistic state-space model with a Gaussian process prior over the transition function and a parametric likelihood is specified by

$$f(\mathbf{x}) \sim \mathcal{GP}\big(m_f(\mathbf{x}), k_f(\mathbf{x}, \mathbf{x}')\big), \tag{2a}$$

$$\mathbf{x}_t \mid \mathbf{f}_t \sim \mathcal{N}(\mathbf{x}_t \mid \mathbf{f}_t, \mathbf{Q}), \tag{2b}$$

$$\mathbf{x}_0 \sim p(\mathbf{x}_0) \tag{2c}$$

$$\mathbf{y}_t \mid \mathbf{x}_t \sim p(\mathbf{y}_t \mid \mathbf{x}_t, \boldsymbol{\theta}_y), \tag{2d}$$

where we have used $\mathbf{f}_t \triangleq f(\mathbf{x}_{t-1})$. Since $f(\mathbf{x}) \in \mathbb{R}^D$, we use the convention that the covariance function $k_f$ returns a $D \times D$ matrix. We group all hyperparameters into $\boldsymbol{\theta} \triangleq \{\boldsymbol{\theta}_f, \boldsymbol{\theta}_y, \mathbf{Q}\}$. Note that

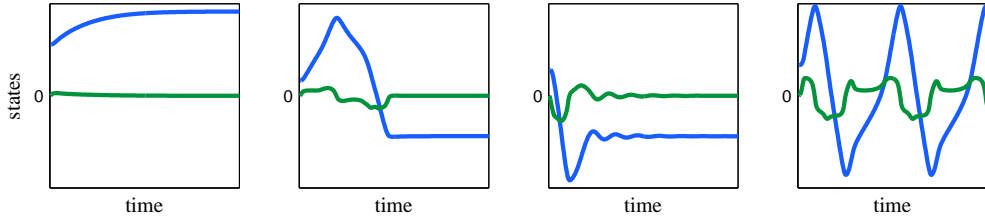

Figure 1: State trajectories from four 2-state nonlinear dynamical systems sampled from a GP-SSM prior with *fixed* hyperparameters. The same prior generates systems with qualitatively different behaviors, e.g. the leftmost panel shows behavior similar to that of a non-oscillatory linear system whereas the rightmost panel appears to have arisen from a limit cycle in a nonlinear system.

we are not restricting the likelihood (2d) to any particular form. The joint distribution of a GP-SSM is

$$p(\mathbf{y}, \mathbf{x}, \mathbf{f}) = p(\mathbf{x}_0) \prod_{t=1}^{T} p(\mathbf{y}_t | \mathbf{x}_t) p(\mathbf{x}_t | \mathbf{f}_t) p(\mathbf{f}_t | \mathbf{f}_{1:t-1}, \mathbf{x}_{0:t-1}), \tag{3}$$

where we use the convention $\mathbf{f}_{1:0} = \emptyset$ and omit the conditioning on $\boldsymbol{\theta}$ in the notation. The GP on the transition function induces a distribution over the latent function values with the form of a GP predictive:

$$p(\mathbf{f}_t | \mathbf{f}_{1:t-1}, \mathbf{x}_{0:t-1}) = \mathcal{N}\big(m_f(\mathbf{x}_{t-1}) + \mathbf{K}_{t-1,0:t-2} \mathbf{K}_{0:t-2,0:t-2}^{-1} (\mathbf{f}_{1:t-1} - m_f(\mathbf{x}_{0:t-2})),$$

$$\mathbf{K}_{t-1,t-1} - \mathbf{K}_{t-1,0:t-2} \mathbf{K}_{0:t-2,0:t-2}^{-1} \mathbf{K}_{t-1,0:t-2}^{\top}\big), \tag{4}$$

where the subindices of the kernel matrices indicate the arguments to the covariance function necessary to build each matrix, e.g. $\mathbf{K}_{t-1,0:t-2} = [k_f(\mathbf{x}_{t-1}, \mathbf{x}_0) \dots k_f(\mathbf{x}_{t-1}, \mathbf{x}_{t-2})]$. When $t = 1$, the distribution is that of a GP marginal $p(\mathbf{f}_1 | \mathbf{x}_0) = \mathcal{N}(m_f(\mathbf{x}_0), k_f(\mathbf{x}_0, \mathbf{x}_0))$.

Equation (3) provides a sequential procedure to sample state trajectories and observations. GP-SSMs are doubly stochastic models in the sense that one could, at least notionally, first sample a state transition dynamics function from eq. (2a) and then, conditioned on that function, sample the state trajectory and observations.

GP-SSMs are a very rich prior over nonlinear dynamical systems. In Fig. 1 we illustrate this concept by showing state trajectories sampled from a GP-SSM with fixed hyperparameters. The dynamical systems associated with each of these trajectories are qualitatively very different from each other. For instance, the leftmost panel shows the dynamics of an almost linear non-oscillatory system whereas the rightmost panel corresponds to a limit cycle in a nonlinear system. Our goal in this paper is to use this prior over dynamical systems and obtain a tractable approximation to the posterior over dynamical systems given the data.

## 3 Variational Inference in GP-SSMs

Since the GP-SSM is a nonparametric model, in order to define a posterior distribution over $f(\mathbf{x})$ and make probabilistic predictions it is necessary to first find the smoothing distribution $p(\mathbf{x}_{0:T} | \mathbf{y}_{1:T})$. Frigola et al. [8] obtained samples from the smoothing distribution that could be used to define a predictive density via Monte Carlo integration. This approach is expensive since it requires averaging over $L$ state trajectory samples of length $T$. In this section we present an alternative approach that aims to find a tractable distribution over the state transition function that is independent of the length of the time series. We achieve this by using variational sparse GP techniques [22].

### 3.1 Augmenting the Model with Inducing Variables

As a first step to perform variational inference in a GP-SSM, we augment the model with $M$ inducing points $\mathbf{u} \triangleq \{\mathbf{u}_i\}_{i=1}^M$. Those inducing points are jointly Gaussian with the latent function values. In the case of a GP-SSM, the joint probability density becomes

$$p(\mathbf{y}, \mathbf{x}, \mathbf{f}, \mathbf{u}) = p(\mathbf{x}, \mathbf{f} | \mathbf{u}) \, p(\mathbf{u}) \prod_{t=1}^{T} p(\mathbf{y}_t | \mathbf{x}_t), \tag{5}$$

where

$$p(\mathbf{u}) = \mathcal{N}(\mathbf{u} \mid \mathbf{0}, \mathbf{K}_{\mathbf{u},\mathbf{u}}) \tag{6a}$$

$$p(\mathbf{x}, \mathbf{f}|\mathbf{u}) = p(\mathbf{x}_0) \prod_{t=1}^{T} p(\mathbf{f}_t|\mathbf{f}_{1:t-1}, \mathbf{x}_{0:t-1}, \mathbf{u})p(\mathbf{x}_t|\mathbf{f}_t), \tag{6b}$$

$$\prod_{t=1}^{T} p(\mathbf{f}_t|\mathbf{f}_{1:t-1}, \mathbf{x}_{0:t-1}, \mathbf{u}) = \mathcal{N}\big(\mathbf{f}_{1:T} \mid \mathbf{K}_{0:T-1,\mathbf{u}}\mathbf{K}_{\mathbf{u},\mathbf{u}}^{-1}\mathbf{u}, \mathbf{K}_{0:T-1} - \mathbf{K}_{0:T-1,\mathbf{u}}\mathbf{K}_{\mathbf{u},\mathbf{u}}^{-1}\mathbf{K}_{0:T-1,\mathbf{u}}^{\top}\big). \tag{6c}$$

Kernel matrices relating to the inducing points depend on a set of inducing inputs $\{\mathbf{z}_i\}_{i=1}^{M}$ in such a way that $\mathbf{K}_{\mathbf{u},\mathbf{u}}$ is an $MD \times MD$ matrix formed with blocks $k_f(\mathbf{z}_i, \mathbf{z}_j)$ having size $D \times D$. For brevity, we use a zero mean function and we omit conditioning on the inducing inputs in the notation.

## 3.2  Evidence Lower Bound of an Augmented GP-SSM

Variational inference [1] is a popular method for approximate Bayesian inference based on making assumptions about the posterior over latent variables that lead to a tractable lower bound on the evidence of the model (sometimes referred to as ELBO). Maximizing this lower bound is equivalent to minimizing the Kullback-Leibler divergence between the approximate posterior and the exact one. Following standard variational inference methodology, [1] we obtain the evidence lower bound of a GP-SSM augmented with inducing points

$$\log p(\mathbf{y}|\boldsymbol{\theta}) \geq \int_{\mathbf{x},\mathbf{f},\mathbf{u}} q(\mathbf{x}, \mathbf{f}, \mathbf{u}) \log \frac{p(\mathbf{u})p(\mathbf{x}_0) \prod_{t=1}^{T} p(\mathbf{f}_t|\mathbf{f}_{1:t-1}, \mathbf{x}_{0:t-1}, \mathbf{u})p(\mathbf{y}_t|\mathbf{x}_t)p(\mathbf{x}_t|\mathbf{f}_t)}{q(\mathbf{x}, \mathbf{f}, \mathbf{u})}. \tag{7}$$

In order to achieve tractability, we use a variational distribution that factorizes as

$$q(\mathbf{x}, \mathbf{f}, \mathbf{u}) = q(\mathbf{u})q(\mathbf{x}) \prod_{t=1}^{T} p(\mathbf{f}_t|\mathbf{f}_{1:t-1}, \mathbf{x}_{0:t-1}, \mathbf{u}), \tag{8}$$

where $q(\mathbf{u})$ and $q(\mathbf{x})$ can take any form but the terms relating to $\mathbf{f}$ are taken to match those of the prior (3). As a consequence, the difficult $p(\mathbf{f}_t|...)$ terms inside the log cancel out and lead to the following lower bound

$$\mathcal{L}(q(\mathbf{u}), q(\mathbf{x}), \boldsymbol{\theta}) = -\mathrm{KL}(q(\mathbf{u})\|p(\mathbf{u})) + \mathcal{H}(q(\mathbf{x})) + \int_{\mathbf{x}} q(\mathbf{x}) \log p(\mathbf{x}_0)$$

$$+ \sum_{t=1}^{T} \left\{ \int_{\mathbf{x},\mathbf{u}} q(\mathbf{x})q(\mathbf{u}) \underbrace{\int_{\mathbf{f}_t} p(\mathbf{f}_t|\mathbf{x}_{t-1}, \mathbf{u}) \log p(\mathbf{x}_t|\mathbf{f}_t) + \int_{\mathbf{x}} q(\mathbf{x}) \log p(\mathbf{y}_t|\mathbf{x}_t)}_{\Phi(\mathbf{x}_t, \mathbf{x}_{t-1}, \mathbf{u})} \right\} \tag{9}$$

where KL denotes the Kullback-Leibler divergence and $\mathcal{H}$ the entropy. The integral with respect to $\mathbf{f}_t$ can be solved analytically: $\Phi(\mathbf{x}_t, \mathbf{x}_{t-1}, \mathbf{u}) = -\frac{1}{2}\mathrm{tr}(\mathbf{Q}^{-1}\mathbf{B}_{t-1}) + \log \mathcal{N}(\mathbf{x}_t|\mathbf{A}_{t-1}\mathbf{u}, \mathbf{Q})$ where $\mathbf{A}_{t-1} = \mathbf{K}_{t-1,\mathbf{u}}\mathbf{K}_{\mathbf{u},\mathbf{u}}^{-1}$, and $\mathbf{B}_{t-1} = \mathbf{K}_{t-1,t-1} - \mathbf{K}_{t-1,\mathbf{u}}\mathbf{K}_{\mathbf{u},\mathbf{u}}^{-1}\mathbf{K}_{\mathbf{u},t-1}$.

As in other variational sparse GP methods, the choice of variational distribution (8) gives the ability to precisely learn the latent function at the locations of the inducing inputs. Away from those locations, the posterior takes the form of the prior conditioned on the inducing variables. By increasing the number of inducing variables, the ELBO can only become tighter [22]. This offers a straightforward trade-off between model capacity and computation cost without increasing the risk of overfitting.

## 3.3  Optimal Variational Distribution for $\mathbf{u}$

The optimal distribution of $q(\mathbf{u})$ can be found by setting to zero the functional derivative of the evidence lower bound with respect to $q(\mathbf{u})$

$$q^*(\mathbf{u}) \propto p(\mathbf{u}) \prod_{t=1}^{T} \exp\{\langle \log \mathcal{N}(\mathbf{x}_t|\mathbf{A}_{t-1}\mathbf{u}, \mathbf{Q}) \rangle_{q(\mathbf{x})}\}, \tag{10}$$

where $\langle \cdot \rangle_{q(\mathbf{x})}$ denotes an expectation with respect to $q(\mathbf{x})$. The optimal variational distribution $q^*(\mathbf{u})$ is, conveniently, a multivariate Gaussian distribution. If, for simplicity of notation, we restrict ourselves to $D = 1$ the natural parameters of the optimal distribution are

$$\boldsymbol{\eta}_1 = Q^{-1} \sum_{t=1}^{T} \langle \mathbf{A}_{t-1}^T x_t \rangle_{q(x_t, x_{t-1})}, \qquad \boldsymbol{\eta}_2 = -\frac{1}{2} \left( \mathbf{K}_{\mathbf{uu}}^{-1} + Q^{-1} \sum_{t=1}^{T} \langle \mathbf{A}_{t-1}^T \mathbf{A}_{t-1} \rangle_{q(x_{t-1})} \right). \quad (11)$$

The mean and covariance matrix of $q^*(\mathbf{u})$, denoted as $\boldsymbol{\mu}$ and $\boldsymbol{\Sigma}$ respectively, can be computed as $\boldsymbol{\mu} = \boldsymbol{\Sigma}\boldsymbol{\eta}_1$ and $\boldsymbol{\Sigma} = (-2\boldsymbol{\eta}_2)^{-1}$. Note that the optimal $q(\mathbf{u})$ depends on the sufficient statistics $\boldsymbol{\Psi}_1 = \sum_{t=1}^{T} \langle \mathbf{K}_{t-1,\mathbf{u}}^T x_t \rangle_{q(x_t, x_{t-1})}$ and $\boldsymbol{\Psi}_2 = \sum_{t=1}^{T} \langle \mathbf{K}_{t-1,\mathbf{u}}^T \mathbf{K}_{t-1,\mathbf{u}} \rangle_{q(x_{t-1})}$.

### 3.4 Optimal Variational Distribution for x

In an analogous way as for $q^*(\mathbf{u})$, we can obtain the optimal form of $q(\mathbf{x})$

$$q^*(\mathbf{x}) \propto p(\mathbf{x}_0) \prod_{t=1}^{T} p(\mathbf{y}_t | \mathbf{x}_t) \exp\{-\frac{1}{2}\mathrm{tr}\big(\mathbf{Q}^{-1}(\mathbf{B}_{t-1} + \mathbf{A}_{t-1}\boldsymbol{\Sigma}\mathbf{A}_{t-1}^T)\big)\} \mathcal{N}(\mathbf{x}_t | \mathbf{A}_{t-1}\boldsymbol{\mu}, \mathbf{Q}), \quad (12)$$

where, in the second equation, we have used $q(\mathbf{u}) = \mathcal{N}(\mathbf{u}|\boldsymbol{\mu}, \boldsymbol{\Sigma})$.

The optimal distribution $q^*(\mathbf{x})$ is equivalent to the smoothing distribution of an auxiliary parametric state-space model. The auxiliary model is simpler than the original one in (3) since the latent states factorize with a Markovian structure. Equation (12) can be interpreted as a nonlinear state-space model with a Gaussian state transition density, $\mathcal{N}(\mathbf{x}_t | \mathbf{A}_{t-1}\boldsymbol{\mu}, \mathbf{Q})$, and a likelihood augmented with an additional term: $\exp\{-\frac{1}{2}\mathrm{tr}\big(\mathbf{Q}^{-1}(\mathbf{B}_{t-1} + \mathbf{A}_{t-1}\boldsymbol{\Sigma}\mathbf{A}_{t-1}^T)\big)\}$.

Smoothing in nonlinear Markovian state-space models is a standard problem in the context of time series modeling. There are various existing strategies to find the smoothing distribution which could be used depending on the characteristics of each particular problem [20]. For instance, in a mildly nonlinear system with Gaussian noise, an extended Kalman smoother can have very good performance. On the other hand, problems with severe nonlinearities and/or non-Gaussian likelihoods can lead to heavily multimodal smoothing distributions that are better represented using particle methods. We note that the application of sequential Monte Carlo (SMC) is particularly straightforward in the present auxiliary model.

### 3.5 Optimizing the Evidence Lower Bound

Algorithm 1 presents a procedure to maximize the evidence lower bound by alternatively sampling from the smoothing distribution and taking steps both in $\boldsymbol{\theta}$ and in the natural parameters of $q^*(\mathbf{u})$. We propose a hybrid variational-sampling approach whereby approximate samples from $q^*(\mathbf{x})$ are obtained with a sequential Monte Carlo smoother. However, as discussed in section 3.4, depending on the characteristics of the dynamical system, other smoothing methods could be more appropriate [20]. As an alternative to smoothing on the auxiliary dynamical system in (12), one could force a $q(\mathbf{x})$ from a particular family of distributions and optimise the evidence lower bound with respect to its variational parameters. For instance, we could posit a Gaussian $q(\mathbf{x})$ with a sparsity pattern in the covariance matrix assuming zero covariance between non-neighboring states and maximize the ELBO with respect to the variational parameters.

We use stochastic gradient descent [10] to maximize the ELBO (where we have plugged in the optimal $q^*(\mathbf{u})$ [22]) by using its gradient with respect to the hyperparameters. Both quantities are stochastic in our hybrid approach due to variance introduced by the sampling of $q^*(\mathbf{x})$. In fact, vanilla sequential Monte Carlo methods will result in biased estimators of the gradient and the parameters of $q^*(\mathbf{u})$. However, in our experiments this has not been an issue. Techniques such as particle MCMC would be a viable alternative to conventional sequential Monte Carlo [13].

**Algorithm 1** Variational learning of GP-SSMs with particle smoothing. Batch mode (i.e. non-SVI) is the particular case where the mini-batch is the whole dataset.

---
**Require:** Observations $\mathbf{y}_{1:T}$. Initial values for $\boldsymbol{\theta}, \boldsymbol{\eta}_1$ and $\boldsymbol{\eta}_2$. Schedules for $\rho$ and $\lambda$. $i = 1$.

  **repeat**
    $\mathbf{y}_{\tau:\tau'} \leftarrow \text{SAMPLEMINIBATCH}(\mathbf{y}_{1:T})$
    $\{\mathbf{x}_{\tau:\tau'}\}_{l=1}^{L} \leftarrow \text{GETSAMPLESOPTIMALQX}(\mathbf{y}_{\tau:\tau'}, \boldsymbol{\theta}, \boldsymbol{\eta}_1, \boldsymbol{\eta}_2)$          sample from eq. (12)
    $\nabla_{\boldsymbol{\theta}} \mathcal{L} \leftarrow \text{GETTHETAGRADIENT}(\{\mathbf{x}_{\tau:\tau'}\}_{l=1}^{L}, \boldsymbol{\theta})$          supp. material
    $\boldsymbol{\eta}_1^*, \boldsymbol{\eta}_2^* \leftarrow \text{GETOPTIMALQU}(\{\mathbf{x}_{\tau:\tau'}\}_{l=1}^{L}, \boldsymbol{\theta})$          eq. (11) or (14)
    $\boldsymbol{\eta}_1 \leftarrow \boldsymbol{\eta}_1 + \rho_i(\boldsymbol{\eta}_1^* - \boldsymbol{\eta}_1)$
    $\boldsymbol{\eta}_2 \leftarrow \boldsymbol{\eta}_2 + \rho_i(\boldsymbol{\eta}_2^* - \boldsymbol{\eta}_2)$
    $\boldsymbol{\theta} \leftarrow \boldsymbol{\theta} + \lambda_i \nabla_{\boldsymbol{\theta}} \mathcal{L}$
    $i \leftarrow i + 1$
  **until** ELBO convergence

---

### 3.6 Making Predictions

One of the most appealing properties of our variational approach to learning GP-SSMs is that the approximate predictive distribution of the state transition function can be cheaply computed

$$p(\mathbf{f}_*|\mathbf{x}_*, \mathbf{y}) = \int_{\mathbf{x}, \mathbf{u}} p(\mathbf{f}_*|\mathbf{x}_*, \mathbf{x}, \mathbf{u})\, p(\mathbf{x}|\mathbf{u}, \mathbf{y})\, p(\mathbf{u}|\mathbf{y}) \approx \int_{\mathbf{x}, \mathbf{u}} p(\mathbf{f}_*|\mathbf{x}_*, \mathbf{u})\, p(\mathbf{x}|\mathbf{u}, \mathbf{y})\, q(\mathbf{u})$$

$$= \int_{\mathbf{u}} p(\mathbf{f}_*|\mathbf{x}_*, \mathbf{u})\, q(\mathbf{u}) = \mathcal{N}(\mathbf{f}_*|\mathbf{A}_* \boldsymbol{\mu}, \mathbf{B}_* + \mathbf{A}_* \boldsymbol{\Sigma} \mathbf{A}_*^{\top}). \tag{13}$$

The derivation in eq. (13) contains two approximations: 1) predictions at new test points are considered to depend only on the inducing variables, and 2) the posterior distribution over $\mathbf{u}$ is approximated by a variational distribution.

After pre-computations, the cost of each prediction is $\mathcal{O}(M)$ for the mean and $\mathcal{O}(M^2)$ for the variance. This contrasts with the $\mathcal{O}(TL)$ and $\mathcal{O}(T^2 L)$ complexity of approaches based on sampling from the smoothing distribution where $p(\mathbf{f}_*|\mathbf{x}_*, \mathbf{y}) = \int_{\mathbf{x}} p(\mathbf{f}_*|\mathbf{x}_*, \mathbf{x})\, p(\mathbf{x}|\mathbf{y})$ is approximated with $L$ samples from $p(\mathbf{x}|\mathbf{y})$ [8]. The variational approach condenses the learning of the latent function on the inducing points $\mathbf{u}$ and does not explicitly need the smoothing distribution $p(\mathbf{x}|\mathbf{y})$ to make predictions.

## 4 Stochastic Variational Inference

Stochastic variational inference (SVI) [10] can be readily applied using our evidence lower bound. When the observed time series is long, it can be expensive to compute $q^*(\mathbf{u})$ or the gradient of $\mathcal{L}$ with respect to the hyperparameters and inducing inputs. Since both $q^*(\mathbf{u})$ and $\frac{\partial \mathcal{L}}{\partial \boldsymbol{\theta}/\mathbf{z}_{1:M}}$ depend linearly on $q(\mathbf{x})$ via sufficient statistics that contain a summation over all elements in the state trajectory, we can obtain unbiased estimates of these sufficient statistics by using one or multiple segments of the sequence that are sampled uniformly at random. However, obtaining $q(\mathbf{x})$ also requires a time complexity of $\mathcal{O}(T)$. Yet, in practice, $q(\mathbf{x})$ can be approximated by running the smoothing algorithm locally around those segments. This can be justified by the fact that in a time series context, the smoothing distribution at a particular time is not largely affected by measurements that are far into the past or the future [20]. The natural parameters of $q^*(\mathbf{u})$ can be estimated by using a portion of the time series of length $S$

$$\boldsymbol{\eta}_1 = Q^{-1} \frac{T}{S} \sum_{t=\tau}^{\tau'} \langle \mathbf{A}_{t-1}^T x_t \rangle_{q(x_t, x_{t-1})}, \quad \boldsymbol{\eta}_2 = -\frac{1}{2} \left( \mathbf{K}_{\mathbf{u}\mathbf{u}}^{-1} + Q^{-1} \frac{T}{S} \sum_{t=\tau}^{\tau'} \langle \mathbf{A}_{t-1}^T \mathbf{A}_{t-1} \rangle_{q(x_{t-1})} \right). \tag{14}$$

## 5 Online Learning

Our variational approach to learn GP-SSMs also leads naturally to an online learning implementation. This is of particular interest in the context of dynamical systems as it is often the case that data arrives in a sequential manner, e.g. a robot learning the dynamics of different objects by interacting

Table 1: Experimental evaluation of 1D nonlinear system. Unless otherwise stated, training times are reported for a dataset with $T = 500$ and test times are given for a test set with $10^5$ data points. All pre-computations independent on test data are performed before timing the "test time". Predictive log likelihoods are the average over the full test set. * our PMCMC code did not use fast updates-downdates of the Cholesky factors during training. This does *not* affect test times.

| | Test RMSE | $\log p(\mathbf{x}_{t+1}^{\text{test}}|\mathbf{x}_t^{\text{test}}, \mathbf{y}_{0:T}^{\text{tr}})$ | Train time | Test time |
|---|---|---|---|---|
| Variational GP-SSM | 1.15 | -1.61 | 2.14 min | 0.14 s |
| Var. GP-SSM (SVI, $T = 10^4$) | 1.07 | -1.47 | 4.12 min | 0.14 s |
| PMCMC GP-SSM [8] | 1.12 | -1.57 | 547 min* | 421 s |
| GP-NARX [17] | 1.46 | -1.90 | 0.22 min | 3.85 s |
| GP-NARX + FITC [17, 18] | 1.47 | -1.90 | 0.17 min | 0.23 s |
| Linear (N4SID, [16]) | 2.35 | -2.30 | 0.01 min | 0.11 s |

with them. Online learning in a Bayesian setting consists in sequential application of Bayes rule whereby the posterior after observing data up to time $t$ becomes the prior at time $t + 1$ [2, 15]. In our case, this involves replacing the prior $p(\mathbf{u}) = \mathcal{N}(\mathbf{u}|\mathbf{0}, \mathbf{K}_{\mathbf{u},\mathbf{u}})$ by the approximate posterior $\mathcal{N}(\mathbf{u}|\boldsymbol{\mu}, \boldsymbol{\Sigma})$ obtained in the previous step. The expressions for the update of the natural parameters of $q^*(\mathbf{u})$ with a new mini batch $\mathbf{y}_{\tau:\tau'}$ are

$$\boldsymbol{\eta}_1' = \boldsymbol{\eta}_1 + Q^{-1} \sum_{t=\tau}^{\tau'} \langle \mathbf{A}_{t-1}^T x_t \rangle_{q(x_t, x_{t-1})}, \quad \boldsymbol{\eta}_2' = \boldsymbol{\eta}_2 - \frac{1}{2} Q^{-1} \sum_{t=\tau}^{\tau'} \langle \mathbf{A}_{t-1}^T \mathbf{A}_{t-1} \rangle_{q(x_{t-1})}. \quad (15)$$

## 6 Experiments

The goal of this section is to showcase the ability of variational GP-SSMs to perform approximate Bayesian learning of nonlinear dynamical systems. In particular, we want to demonstrate: 1) the ability to learn the inherent nonlinear dynamics of a system, 2) the application in cases where the latent states have higher dimensionality than the observations, and 3) the use of non-Gaussian likelihoods.

### 6.1 1D Nonlinear System

We apply our variational learning procedure presented above to the one-dimensional nonlinear system described by $p(x_{t+1}|x_t) = \mathcal{N}(f(x_t), 1)$ and $p(y_t|x_t) = \mathcal{N}(x_t, 1)$ where the transition function is $x_t + 1$ if $x < 4$ and $-4x_t + 21$ if $x \geq 4$. Its pronounced kink makes it challenging to learn. Our goal is to find a posterior distribution over this function using a GP-SSM with Matérn covariance function. To solve the expectations with respect to the approximate smoothing distribution $q(\mathbf{x})$ we use a bootstrap particle fixed-lag smoother with 1000 particles and a lag of 10.

In Table 1, we compare our method (Variational GP-SSM) against the PMCMC sampling procedure from [8] taking 100 samples and 10 burn in samples. As in [8], the sampling exhibited very good mixing with 20 particles. We also compare to an auto-regressive model based on Gaussian process regression [17] of order 5 with Matérn ARD covariance function with and without FITC approximation. Finally, we use a linear subspace identification method (N4SID, [16]) as a baseline for comparison. The PMCMC training offers the best test performance from all methods using 500 training points at the cost of substantial train and test time. However, if more data is available ($T = 10^4$) the stochastic variational inference procedure can be very attractive since it improves test performance while having a test time that is independent of the training set size. The reported SVI performance has been obtained with mini-batches of 100 time-steps.

### 6.2 Neural Spike Train Recordings

We now turn to the use of SSMs to learn a simple model of neural activity in rats' hippocampus. We use data in neuron cluster 1 (the most active) from experiment ec013.717 in [14]. In some regions of the time series, the action potential spikes show a clear pattern where periods of rapid spiking are followed by periods of very little spiking. We wish to model this behaviour as an autonomous nonlinear dynamical system (i.e. one not driven by external inputs). Many parametric models of nonlinear neuron dynamics have been proposed [11] but our goal here is to learn a model from data

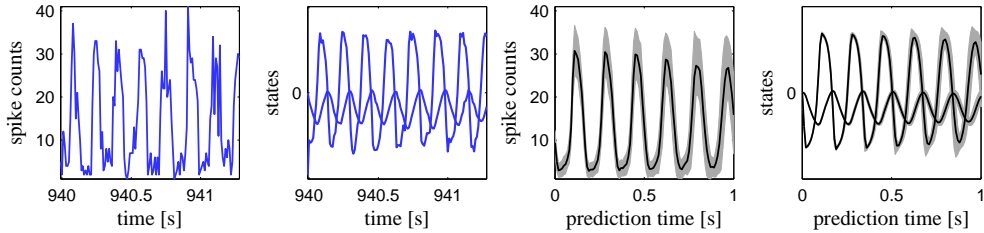

Figure 2: From left to right: 1) part of the observed spike count data, 2) sample from the corresponding smoothing distribution, 3) predictive distribution of spike counts obtained by simulating the posterior dynamical from an initial state, and 4) corresponding latent states.

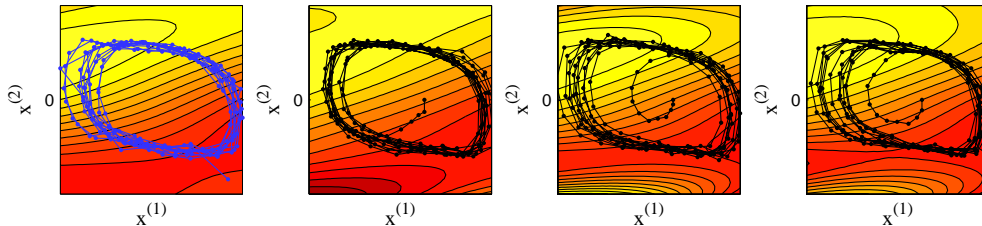

Figure 3: Contour plots of the state transition function $\mathbf{x}_{t+1}^{(2)} = f(\mathbf{x}_t^{(1)}, \mathbf{x}_t^{(2)})$, and trajectories in state space. Left: mean posterior function and trajectory from smoothing distribution. Other three panels: transition functions sampled from the posterior and trajectories simulated conditioned on the corresponding sample. Those simulated trajectories start inside the limit cycle and are naturally attracted towards it. Note how function samples are very similar in the region of the limit cycle.

without using any biological insight. We use a GP-SSM with a structure such that it is the discrete-time analog of a second order nonlinear ordinary differential equation: two states one of which is the derivative of the other. The observations are spike counts in temporal bins of 0.01 second width. We use a Poisson likelihood relating the spike counts to the second latent state $y_t | \mathbf{x}_t \sim$ Poisson$(\exp(\alpha \mathbf{x}_t^{(2)} + \beta))$.

We find a posterior distribution for the state transition function using our variational GP-SSM approach. Smoothing is done with a fixed-lag particle smoother and training until convergence takes approximately 50 iterations of Algorithm 1. Figure 2 shows a part of the raw data together with an approximate sample from the smoothing distribution during the same time interval. In addition, we show the distribution over predictions made by chaining 1-step-ahead predictions. To make those predictions we have switched off process noise ($\mathbf{Q} = \mathbf{0}$) to show more clearly the effect of uncertainty in the state transition function. Note how the frequency of roughly 6 Hz present in the data is well captured. Figure 3 shows how the limit cycle corresponding to a nonlinear dynamical system has been captured (see caption for details).

## 7 Discussion and Future Work

We have derived a tractable variational formulation to learn GP-SSMs: an important class of models of nonlinear dynamical systems that is particularly suited to applications where a principled parametric model of the dynamics is not available. Our approach makes it possible to learn very expressive models without risk of overfitting. In contrast to previous approaches [4, 12, 25], we have demonstrated the ability to learn a nonlinear state transition function in a latent space of greater dimensionality than the observation space. More crucially, our approach yields a tractable posterior over nonlinear systems that, as opposed to those based on sampling from the smoothing distribution [8], results in a computation time for the predictions that does not depend on the length of the time series.

Given the interesting capabilities of variational GP-SSMs, we believe that future work is warranted. In particular, we want to focus on structured variational distributions $q(\mathbf{x})$ that could eliminate the need to solve the smoothing problem in the auxiliary dynamical system at the cost of having more variational parameters to optimize. On a more theoretical side, we would like to better characterize GP-SSM priors in terms of their dynamical system properties: stability, equilibria, limit cycles, etc.

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
