[Supplementary Material]

# Supplementary Material

## A    Remark on $p(\mathbf{f}|\mathbf{x})$

Note that

$$\prod_{t=1}^{T} p(\mathbf{f}_t|\mathbf{f}_{1:t-1}, \mathbf{x}_{0:t-1}, \mathbf{u}) \neq p(\mathbf{f}_{1:T}|\mathbf{x}_{0:T-1}, \mathbf{u}). \tag{16}$$

The right hand side is equivalent to Gaussian process regression where $\mathbf{x}_{0:T-1}$ are inputs and $\mathbf{x}_{1:T}$ are outputs

$$p(\mathbf{f}_{1:T}|\mathbf{x}_{0:T-1}, \mathbf{u}) = \mathcal{N}\big(\mathbf{K}_{0:T-1,\{\mathbf{u},0:T-1\}}(\mathbf{K}_{\{\mathbf{u},0:T-1\}} + \boldsymbol{\Sigma}_{\mathbf{Q}})^{-1} \begin{pmatrix} \mathbf{u} \\ \mathbf{x}_{0:T-1} \end{pmatrix},$$

$$\mathbf{K}_{0:T-1} - \mathbf{K}_{0:T-1,\{\mathbf{u},0:T-1\}}(\mathbf{K}_{\{\mathbf{u},0:T-1\}} + \boldsymbol{\Sigma}_{\mathbf{Q}})^{-1}\mathbf{K}_{0:T-1,\{\mathbf{u},0:T-1\}}^{\top}\big) \tag{17}$$

where $\boldsymbol{\Sigma}_{\mathbf{Q}} = \mathrm{blockdiag}(\mathbf{0}, \mathbf{I} \otimes \mathbf{Q})$ captures process noise.

However, the left hand side of (16) is the product of terms that are equivalent to Gaussian process prediction with noiseless observations

$$p(\mathbf{f}_t|\mathbf{f}_{1:t-1}, \mathbf{x}_{0:t-1}, \mathbf{u}) = \mathcal{N}\big(\mathbf{K}_{t-1,\{\mathbf{u},0:t-2\}}\mathbf{K}_{\{\mathbf{u},0:t-2\}}^{-1} \begin{pmatrix} \mathbf{u} \\ \mathbf{f}_{1:t-1} \end{pmatrix},$$

$$\mathbf{K}_{t-1} - \mathbf{K}_{t-1,\{\mathbf{u},0:t-2\}}\mathbf{K}_{\{\mathbf{u},0:t-2\}}^{-1}\mathbf{K}_{t-1,\{\mathbf{u},0:t-2\}}^{\top}\big). \tag{18}$$

The product of these terms can be succinctly represented by the following Gaussian

$$\prod_{t=1}^{T} p(\mathbf{f}_t|\mathbf{f}_{1:t-1}, \mathbf{x}_{0:t-1}, \mathbf{u}) = \mathcal{N}\big(\mathbf{f}_{1:T} \mid \mathbf{K}_{0:T-1,\mathbf{u}}\mathbf{K}_{\mathbf{u},\mathbf{u}}^{-1}\mathbf{u},$$

$$\mathbf{K}_{0:T-1} - \mathbf{K}_{0:T-1,\mathbf{u}}\mathbf{K}_{\mathbf{u},\mathbf{u}}^{-1}\mathbf{K}_{0:T-1,\mathbf{u}}^{\top}\big). \tag{19}$$

Therefore

$$\int_{\mathbf{f}_t} \prod_{t=1}^{T} p(\mathbf{f}_t|\mathbf{f}_{1:t-1}, \mathbf{x}_{0:t-1}, \mathbf{u}) = \mathcal{N}\big(\mathbf{f}_t \mid \mathbf{K}_{t-1,\mathbf{u}}\mathbf{K}_{\mathbf{u},\mathbf{u}}^{-1}\mathbf{u},$$

$$\mathbf{K}_{t-1} - \mathbf{K}_{t-1,\mathbf{u}}\mathbf{K}_{\mathbf{u},\mathbf{u}}^{-1}\mathbf{K}_{t-1,\mathbf{u}}^{\top}\big). \tag{20}$$

## B    Relationship of Variational Approximation with Other Models

### B.1    Markovian Model with Heteroscedastic Noise

Although not strictly a GP-SSM, it is also interesting to consider a model where the state transitions are independent of each other given the inducing variables

$$p(\mathbf{f}_{1:T}, \mathbf{x}_{0:T}|\mathbf{u}) = p(\mathbf{x}_0) \prod_{t=1}^{T} \mathcal{N}\big(\mathbf{f}_t|\mathbf{A}_{t-1}\mathbf{u}, \mathbf{B}_{t-1}\big)\, p(\mathbf{x}_t|\mathbf{f}_t). \tag{21}$$

This model can be interpreted as a parametric model where $\mathbf{A}_{t-1}\mathbf{u}$ is a deterministic transition function and $\mathbf{B}_{t-1}$ provides the description of an heteroscedastic process noise. Note that this process noise is independent between any two time steps.

If we look for an evidence lower bound with using an analogous procedure to that of section 3 and the following variational distribution

$$q(\mathbf{x}, \mathbf{f}, \mathbf{u}) = q(\mathbf{u})q(\mathbf{x}) \prod_{t=1}^{T} \mathcal{N}\big(\mathbf{f}_t|\mathbf{A}_{t-1}\mathbf{u}, \mathbf{B}_{t-1}\big), \tag{22}$$

the lower bound becomes the same as the one in equation (9).

## B.2 Bayesian RBF Model

The Radial Basis Function in this model uses a deterministic state transition function of the form $f(\mathbf{x}) = \mathbf{A}(\mathbf{x})\,\mathbf{u}$ which leads to

$$p(\mathbf{f}_{1:T}, \mathbf{x}_{0:T}|\mathbf{u}) = p(\mathbf{x}_0)\prod_{t=1}^{T}\mathcal{N}\big(\mathbf{f}_t|\mathbf{A}_{t-1}\,\mathbf{u}, \mathbf{0}\big)\,p(\mathbf{x}_t|\mathbf{f}_t). \tag{23}$$

As opposed to the model in B.1, this transition dynamics is not heteroscedastic. It is fully determin-istic and parameterized by $\mathbf{u}$ and $\mathbf{z}$. From a GP perspective, this model is analogous to the Subset of Regressors sparse GP [18]. This model has the undesirable characteristic that the predictive vari-ance shrinks to zero away from the inducing inputs when exactly the opposite behaviour would be desirable.

## B.3 Double GP Model

A state space model having a Gaussian process prior over the state transition function *and* the emis-sion/observation function can be represented by

$$f(\mathbf{x}) \sim \mathcal{GP}\big(m_f(\mathbf{x}), k_f(\mathbf{x}, \mathbf{x}')\big), \tag{24a}$$

$$g(\mathbf{x}) \sim \mathcal{GP}\big(m_g(\mathbf{x}), k_g(\mathbf{x}, \mathbf{x}')\big), \tag{24b}$$

$$\mathbf{x}_0 \sim p(\mathbf{x}_0) \tag{24c}$$

$$\mathbf{x}_t \mid \mathbf{f}_t \sim \mathcal{N}(\mathbf{x}_t \mid \mathbf{f}_t, \mathbf{Q}), \tag{24d}$$

$$\mathbf{y}_t \mid \mathbf{g}_t \sim \mathcal{N}(\mathbf{y}_t \mid \mathbf{g}_t, \mathbf{R}), \tag{24e}$$

where we have used $\mathbf{f}_t \triangleq f(\mathbf{x}_{t-1})$ and $\mathbf{g}_t \triangleq g(\mathbf{x}_t)$. If the transition GP is augmented with inducing variables $\mathbf{u}$ and the emission GP is augmented with $\mathbf{v}$, we obtain the following joint distribution of the model

$$p(\mathbf{y}, \mathbf{x}, \mathbf{f}, \mathbf{u}, \mathbf{g}, \mathbf{v}) = p(\mathbf{g}|\mathbf{x}, \mathbf{v})\,p(\mathbf{x}, \mathbf{f}|\mathbf{u})\,p(\mathbf{u})\,p(\mathbf{v})\prod_{t=1}^{T}p(\mathbf{y}_t|\mathbf{g}_t), \tag{25}$$

where $p(\mathbf{x}, \mathbf{f}|\mathbf{u})$ is the same as in the model presented in the paper and $p(\mathbf{g}|\mathbf{x}, \mathbf{v})$ is straightforward since it is conditioned on all the states.

We use the following variational distribution over latent variables

$$q(\mathbf{x}, \mathbf{f}, \mathbf{u}, \mathbf{g}, \mathbf{v}) = q(\mathbf{u})q(\mathbf{v})q(\mathbf{x})p(\mathbf{g}|\mathbf{x}, \mathbf{v})\prod_{t=1}^{T}p(\mathbf{f}_t|\mathbf{f}_{1:t-1}, \mathbf{x}_{0:t-1}, \mathbf{u}). \tag{26}$$

Terms with latent variables inside kernel matrices cancel inside the log

$$\begin{aligned}
\log p(\mathbf{y}|\boldsymbol{\theta}) \geq & \int_{\mathbf{x}, \mathbf{f}, \mathbf{u}, \mathbf{g}, \mathbf{v}} q(\mathbf{x}, \mathbf{f}, \mathbf{u}, \mathbf{g}, \mathbf{v}) \log \frac{p(\mathbf{u})p(\mathbf{v})p(\mathbf{x}_0)\prod_{t=1}^{T}p(\mathbf{y}_t|\mathbf{g}_t)p(\mathbf{x}_t|\mathbf{f}_t)}{q(\mathbf{u})q(\mathbf{v})q(\mathbf{x})} \\
= & -\mathrm{KL}(q(\mathbf{u})\|p(\mathbf{u})) - \mathrm{KL}(q(\mathbf{v})\|p(\mathbf{v})) + \mathcal{H}(q(\mathbf{x})) + \int_{\mathbf{x}} q(\mathbf{x})\log p(\mathbf{x}_0) \\
& + \sum_{t=1}^{T}\bigg\{ \int_{\mathbf{x}, \mathbf{u}} q(\mathbf{x})q(\mathbf{u}) \int_{\mathbf{f}_t} p(\mathbf{f}_t|\mathbf{x}_{t-1}, \mathbf{u}) \log p(\mathbf{x}_t|\mathbf{f}_t) \\
& + \int_{\mathbf{x}, \mathbf{v}} q(\mathbf{x})q(\mathbf{v}) \int_{\mathbf{g}_t} p(\mathbf{g}_t|\mathbf{x}_t, \mathbf{v}) \log p(\mathbf{y}_t|\mathbf{g}_t)\bigg\}.
\end{aligned}$$

The optimal distribution $q^*(\mathbf{u})$ is the same as in eq. (10) and the optimal variational distribution of the emission inducing variables is a Gaussian distribution

$$q^*(\mathbf{v}) \propto p(\mathbf{v})\prod_{t=1}^{T}\exp\{\langle\log\mathcal{N}(\mathbf{y}_t|\mathbf{C}_t\,\mathbf{v}, \mathbf{R})\rangle_{q(\mathbf{x}_t)}\} \tag{27}$$

where

$$\mathbf{C}_t = \mathbf{K}_{t,\mathbf{v}}\mathbf{K}_{\mathbf{v},\mathbf{v}}^{-1},$$
$$\mathbf{D}_t = \mathbf{K}_{t,t} - \mathbf{K}_{t,\mathbf{v}}\mathbf{K}_{\mathbf{v},\mathbf{v}}^{-1}\mathbf{K}_{\mathbf{v},t}.$$

The optimal variational distribution of the state trajectory is

$$q^*(\mathbf{x}) \propto p(\mathbf{x}_0) \prod_{t=1}^{T} \exp\{-\frac{1}{2}\mathrm{tr}(\mathbf{Q}^{-1}(\mathbf{B}_{t-1} + \mathbf{A}_{t-1}\boldsymbol{\Sigma}\mathbf{A}_{t-1}^{T})) - \frac{1}{2}\mathrm{tr}(\mathbf{R}^{-1}(\mathbf{D}_t + \mathbf{C}_t\boldsymbol{\Lambda}\mathbf{C}_t^{T}))\}$$
$$\mathcal{N}(\mathbf{x}_t|\mathbf{A}_{t-1}\boldsymbol{\mu},\mathbf{Q})\,\mathcal{N}(\mathbf{y}_t|\mathbf{C}_t\boldsymbol{\nu},\mathbf{R}), \tag{28}$$

where we have used $q(\mathbf{v}) = \mathcal{N}(\boldsymbol{\nu},\boldsymbol{\Lambda})$.

## C  Optimization of Hyperparameters

We optimize the hyperparameters and variational parameters ($\mathbf{z}_{1:M}$) with gradient ascent. The gradient w.r.t. $\boldsymbol{\theta}$ is computed as

$$\frac{\partial\mathcal{L}}{\partial\boldsymbol{\theta}} = \left\langle \frac{\partial}{\partial\boldsymbol{\theta}}\log p(\mathbf{u}) \right\rangle_{q(\mathbf{u})} + \left\langle \frac{\partial}{\partial\boldsymbol{\theta}}\log p(\mathbf{x}_0) \right\rangle_{q(\mathbf{x}_0)}$$
$$+ \sum_{t=1}^{T} \left\{ \left\langle -\frac{1}{2}\frac{\partial}{\partial\boldsymbol{\theta}}\mathbf{tr}(\mathbf{Q}^{-1}(\mathbf{B}_{t-1} + \mathbf{A}_{t-1}\boldsymbol{\Sigma}\mathbf{A}_{t-1}^{T})) \right\rangle_{q(\mathbf{x}_{t-1})} + \left\langle \frac{\partial}{\partial\boldsymbol{\theta}}\log\mathcal{N}(\mathbf{x}_t|\mathbf{A}_{t-1}\boldsymbol{\mu},\mathbf{Q}) \right\rangle_{q(\mathbf{x}_t,\mathbf{x}_{t-1})}$$
$$+ \left\langle \frac{\partial}{\partial\boldsymbol{\theta}}\log p(\mathbf{y}_t|\mathbf{x}_t) \right\rangle_{q(\mathbf{x}_t)} \right\}. \tag{29}$$

The gradient with respect to $\mathbf{z}_{1:M}$ is similar with $\boldsymbol{\theta}$ replaced by $\mathbf{z}_{1:M}$. In this expression $\boldsymbol{\mu}$ and $\boldsymbol{\Sigma}$ can be replaced by their optimal settings dependent on the sufficient statistics $\boldsymbol{\Psi}_1$ and $\boldsymbol{\Psi}_2$.

## D  Plots from Experiments Section

Figure 4: Posterior distribution over latent state transition function (green: ground truth, blue: posterior mean, red: mean $\pm 1$ standard deviation).