[Reviews · NeurIPS 2014]

Submitted by Assigned_Reviewer_3

The paper presents a VB method for learning nonlinear state-space models using sparse GPs to model the nonlinear state transition and observation mappings. The proposed method looks very good and efficient, but the empirical evaluation is relatively weak.

Quality: The paper appears technically sound, save one minor problem listed below. The method is based on existing solid principles with VB-based sparse GPs, stochastic variational inference and sequential Monte Carlo. The experimental evaluation is limited to two relatively small systems and includes no comparisons to other methods. Explicit experimental comparisons with existing nonlinear SSM learning methods would strengthen the paper.

Clarity: The paper is in principle well-written and easy to read. A lot of details are missing, especially about the SMC but also about other details such as number and selection of the inducing inputs as well as more details of the covariance functions. Including all possible details in the paper is especially important as there is no code available for the method.

Originality: The proposed method is novel. The differences of the method and previous work are explained clearly and relevant earlier work is cited properly.

Significance: Learning models of nonlinear dynamical systems is an important problem with many potential applications. The paper promises significant improvement in efficiency over previous approaches in this problem. If there were a nice implementation available, I believe it could have quite significant impact. As is, without an implementation the direct practical impact is more uncertain because of the complexity of the method and its implementation, but it will still likely be useful as a basis of future research.

Specific comments:
Eq. (4): Shouldn't you have (f_{1:t-1}-mu_f_{1:t-1}) in the mean instead of just f_{1:t-1}?

Line 262-264: I believe a sparse Gaussian MRF with a sparse precision matrix (zeros between non-neighbouring states) would make more sense than a sparse covariance matrix.

Line 268: The comment on biasedness of "vanilla" SMC estimators suggests that the question of sampling q^*(x) should probably be discussed in more detail. The current text is quite confusing as on one hand it says the method uses SMC without specifying any details, and next it says SMC is biased. For someone aiming to implement the method, it would be valuable to discuss this in more detail to know what is valid and what is not.

Sec. 6.1-6.2: Which mean functions were used in the experiments?

Sec. 6.2: which covariance function was used? How big was the data set?

Update after author feedback: the new experimental results presented in feedback clearly strengthen the argument of the paper. The authors seem to have avoided the comments about clarity and lack of details - I really hope they will take those into account in the final submission if the paper is accepted.
Summary: A solid paper providing an apparently efficient solution to learning a nonlinear state space model with GPs. Could be improved by making the presentation and the method more easily accessible, and adding more comprehensive empirical evaluation, but makes a nice contribution as is.

Submitted by Assigned_Reviewer_10

This paper presents a variational algorithm for non-linear state-space model based on Gaussian processes. In the earlier work (Frigola et al.), expensive sampling-based approach was taken to sample from the smoothing distribution, while a different approach, which augments the model with inducing points, is presented. This approach provides a tractable posterior over the non-liner dynamical systems. It reduces the computational burden, in the meantime provides a trade off between model capacity and computational cost. The extensions to stochastic variational inference and online learning are presented as well. Experimental results on synthetic data and real-world neural spike data are reported, which demonstrate the effectiveness of the proposed algorithm.

Overall, the paper is well-written with clear motivation and connects naturally with the previous work in this subject. I am not a GP expert, but the paper does provide a clear path of related literatures and make a good point about why the problem being solved is essential. I have a couple of minor comments regarding the experiments:

1. The paper mentioned a couple times that the variational framework provides a knob (inducing points) to trade off between model capacity and computation time. Though theoretically justified, it would be better to demonstrate it experimentally.

2. Also, it would be interesting to compare the variational framework with the Particle MCMC proposed in Frigola et al. on a relatively shorter time series (so that computation time is not insane for the sample-based approach) to see how much is lost by the variational approximation - especially the choice of q(f) being the prior p(f|..) seems to be quite a rough approximation. (Line 191, shouldn’t it be Equation (6c) instead of (3)?)

Summary: This paper presents a variational framework for inference in non-linear state-space model based on Gaussian processes. By augmenting the model with inducing points, the inference can be carried out independent of the length of time series, which is more efficient compared to earlier sampling based approach, also a tractable approximated posterior over the dynamical systems can be obtained. The paper is well-written, therefore I vote for acceptance.

Submitted by Assigned_Reviewer_32

The paper is concerned with learning of non-linear dynamics in non-linear stochastic state-space models via the use of sparse GPs. The basic approach is to put Gaussian prior on the transition function in the state-space model, which as such is a simple idea applied many times before. However, inferring GP and the state from observations is computationally intractable problem and therefore approximations are needed. This problem is indeed timely and important.

The authors construct the tractable approximation by using a set of inducing points and a variational approximation based inference method. The variational approximation is chosen suitably so that certain difficult terms cancel and the approximating distributions q(u) turns out to be Gaussian. Furthermore, the other approximating distribution q(x) turns out to have a form which is computable using state-space smoothing methods. The authors use particle smoothing for this. As an additional approximation the authors propose a stochastic approximation for the optimisation of the variational lower bound and optimal learning version of the method.

A disappointment is that the approach is indeed a combination of known approaches and, for example, the use of inducing points in SSM/GP models is not a new idea either. Furthermore, using inducing points essentially approximates the GP with a parametric model which, as the authors also mention, is an ancient idea. However, a good thing is that the particular variational approximation indeed seems to be novel small idea and has its benefits. In particular, an approximate dynamic model prediction can be done cheaply and it allows for the use of various methods from state-space model context (including particle smoothers). The experiments indeed illustrate the approach well.

Due to the use of inducing points, I would expect that the scaling with respect to the state dimensionality is bad (= exponential). This could be discussed. It would also be useful to know the actual CPU times needed in the experiments. In the second experiment you could also document the number of inducing points and their locations.

Minor points:

- You use "e.g." in various places -- you shouldn't use abbreviations in scientific text.
- All the dx's and du's are missing in the integrals (13).
- In 6.1 the notation like p(y_t | x_t) = N(x_t, 1) is somewhat clumsy, because the right hand side doesn't depend on y_t at all. y_t | x_t ~ N(x_t, 1) should be fine or p(y_t | x_t) = N(y_t | x_t, 1).
- Should x' and Q's in equations (11) and (14) [and below (11)] be in bold face?
Summary: A somewhat novel approach to an important computationally intractable problem. Numerical experiments are good.
Author Feedback
Author rebuttal: We thank the reviewers for their very encouraging feedback. In order to further strengthen the paper, we have run experiments comparing our algorithm to particle-based methods, auto-regressive models based on Gaussian process and, as a sanity check, linear models. These experiments have been run on the data from section 6.1 in the paper. A summary of the results can be found here:

http://i.imgur.com/mE99KR9.png

The results of these experiments are quite interesting. The sampling procedure from [8] establishes itself as the best performing method. However, it does so at a significant computational cost and its performance margin is small. The experiments also show that when relatively large datasets are available, stochastic variational inference provides very good performance at a reduced training cost and a test time that is independent of the size of the training set.

We believe that these experiments have strengthened the paper and that variational inference on GP-SSMs has the potential to become a useful tool to learn models of complex nonlinear dynamical systems from large amounts of data. This has wide applicability outside the field of machine learning and our next challenge is to package the algorithm in such a way that can be used successfully by non-specialists.